# PPE51 Is Involved in the Uptake of Disaccharides by *Mycobacterium tuberculosis*

**DOI:** 10.3390/cells9030603

**Published:** 2020-03-03

**Authors:** Małgorzata Korycka-Machała, Jakub Pawełczyk, Paulina Borówka, Bożena Dziadek, Anna Brzostek, Malwina Kawka, Adrian Bekier, Sebastian Rykowski, Agnieszka B. Olejniczak, Dominik Strapagiel, Zbigniew Witczak, Jarosław Dziadek

**Affiliations:** 1Institute of Medical Biology, Polish Academy of Sciences, 93-232 Lodz, Poland; mkorycka@cbm.pan.pl (M.K.-M.); jpawelczyk@cbm.pan.pl (J.P.); abrzostek@cbm.pan.pl (A.B.); srykowski@cbm.pan.pl (S.R.); aolejniczak@cbm.pan.pl (A.B.O.); 2Biobank Lab, Department of Molecular Biophysics, Faculty of Biology and Environmental Protection, University of Lodz, 90-237 Lodz, Poland; paulina.borowka@biol.uni.lodz.pl (P.B.); dominik.strapagiel@biol.uni.lodz.pl (D.S.); 3Department of Anthropology, Faculty of Biology and Environmental Protection, University of Lodz, 90-237 Lodz, Poland; 4Department of Immunoparasitology, Faculty of Biology and Environmental Protection, University of Lodz, 90-237 Lodz, Poland; bozena.dziadek@biol.uni.lodz.pl (B.D.); malwina.kawka@biol.uni.lodz.pl (M.K.); adrian.bekier@unilodz.eu (A.B.); 5Department of Pharmaceutical Sciences, Nesbitt School of Pharmacy, Wilkes University, Wilkes-Barre, PA 18766, USA; zbigniew.witczak@wilkes.edu

**Keywords:** PPE51, tuberculosis, disaccharides, *thio*-sugars, nutrient uptake

## Abstract

We have recently found that selected *thio*-disaccharides possess bactericidal effects against *Mycobacterium tuberculosis* but not against *Escherichia coli* or S*taphylococcus aureus*. Here, we selected spontaneous mutants displaying resistance against the investigated *thio*-glycoside. According to next-generation sequencing, four of six analyzed mutants which were resistant to high concentrations of the tested chemical carried nonsynonymous mutations in the gene encoding the PPE51 protein. The complementation of these mutants with an intact *ppe51* gene returned their sensitivity to the wild-type level. The uptake of tritiated *thio*-glycoside was significantly more abundant in wild-type *Mycobacterium tuberculosis* compared to the strain carrying the mutated *ppe51* gene. The *ppe51* mutations or CRISPR-Cas9-mediated downregulation of PPE51 expression affected the growth of mutant strains on minimal media supplemented with disaccharides (maltose or lactose) but not with glycerol or glucose as the sole carbon and energy source. Taking the above into account, we postulate that PPE51 participates in the uptake of disaccharides by tubercle bacilli.

## 1. Introduction

Tuberculosis, caused by *Mycobacterium tuberculosis* (*Mtb*), is among the most serious bacterial infectious diseases of the modern world. The World Health Organization [1] estimates that, each year, approximately ten million people fall ill with the disease and 1.5 million people die. Tuberculosis therapy lasts 6 to 24 months, depending on the drug susceptibility of the infecting bacterial strain and its metabolic state. Particularly, the treatment of multi-drug-resistant tuberculosis (MDR-TB) is very expensive, very difficult for patients to follow, carries a burden of side effects and is successful only in about 54% [2,3]. As determined by the sequencing of the first *Mtb* genome, 7% of its total coding potential is dedicated to the PE and PPE families, which are characterized by the presence of a conserved Pro-Glu (PE) and Pro-Pro-Glu (PPE) motif in their 110 and 180 amino acid N-terminal regions, respectively [4]. On the other hand, the C-termini of PE and PPE remain highly variable [5]. According to their C-terminal domains, the PE and PPE families are further divided into subgroups, as recently described in an excellent review by [6]. The genome of the laboratory strain *Mtb* H_37_Rv contains 99 *pe* and 69 *ppe* genes, but these numbers are variable in different clinical strains [7,8]. Moreover, *pe*/*ppe* genes are often identified in pairs and are likely co-expressed; however, the individual genes are also present throughout the genome [9]. PE and PPE are not found outside of the genus *Mycobacterium*, and they are largely found in slow-growing species, with fast-growing mycobacteria possessing far fewer *pe/ppe* genes; only two pairs of PE and PPE proteins have been identified in *M. smegmatis* [8].

The PE and PPE proteins have evolved in association with the duplication of *esx* gene cluster regions encoding type VII secretion systems [8]. The ESX system seems to have a significant role in the export of PE/PPE proteins, where ESX-5 is responsible for exporting various proteins lacking Sec or Tat signal peptides [10,11]. The crystal structure of a PE-PPE heterodimer bound to ESX secretion-associated protein G (EspG) showed the interaction between EspG and the PPE domain [12]. It was proposed that EspG delivers PE-PPE to ESX machinery for secretion, and the secretion of most PE-PPE proteins in *Mtb* is mediated by EspG from the ESX-5 system [13]. ESX-5 is only present in slow-growing mycobacteria and is responsible for the secretion of multiple substrates. As identified by proteomic analysis, all detectable PE and PPE proteins in the cell surface and cell envelope fractions are routed through ESX-5 [14]. The growth analysis of the *M. marinum esx5* mutant on defined carbon sources revealed that ESX-5 is involved in the uptake of fatty acids. Ates et al. [14] postulated that the ESX-5 system is responsible for the transport of cell envelope proteins that are required for nutrient uptake. Authors have speculated that these proteins might in this way compensate for the lack of MspA-like porins in slow-growing mycobacteria.

Mitra et al. [15] identified PPE36 and PPE62, as well as Rv0265c as heme-binding cell surface receptors of *Mtb* essential for heme utilization. More recently, Tullius et al. [16] identified PPE37 as being essential for heme-iron acquisition in *Mtb*. Surprisingly, a later paper did not confirm the role of PPE36 in heme utilization; however, the authors of both papers used different *Mtb* strains and media for their work.

It was reported that some PPE proteins are secreted to the bacterial surface and interact with other proteins, as well as components of the host immune system [17,18]. PPE proteins affect host–pathogen interactions and immune evasion [19]. PPE-dependent immune escape during *Mtb* infection [20] the interactions of PPE with toll-like receptor 2 (TLR-2), cytokine release by activation of macrophages and dendritic cells, promoting apoptosis and necrosis of host cells, have also been reported [21,22,23,24,25]. PE/PPE proteins need to be surface-associated or released in a soluble form to interact directly with the host; therefore, they require transportation throughout the bacterial inner membrane with their cognate Type VII secretion systems, ESX-1, ESX-3 and ESX-5 [26,27,28,29]. The ESX-3 secretion system is involved in the transport of PPE-PPW proteins engaged in iron acquisition throughout mycobactin (PPE4/PE5) or heme (PPE36/PPE37) [15,16,30,31]. On the other hand, PPE-SVP and PPE-MPTR are secreted by ESX-5 [10,14,32]. PE8/PPE15 of the PPE-SVP family compose an operon together with EsxI-J and are necessary for the secretion of a specific subset of proteins that are important for bacterial virulence in *Mtb* and *M. marinum* [33]. PPE38 is required for the secretion of all detected PE_PGRS and PPE-MPTR proteins [34].

PE/PPE proteins are important players in host–pathogen interactions and affect the immunological response of the host organism. On the other hand, some members of the PE/PPE family are outer membrane nutrient transport proteins involved in iron acquisition.

Numerous studies have established that *Mtb* relies on fatty acids and cholesterol in a nutritionally stringent environment of the macrophage phagolysosome during latency [35]. However, during its life-cycle in the necrotic tissue and caseum or the lymph and blood of a newly infected person, *Mtb* also encounters a transient abundance of other carbon sources such as carbohydrates [36,37]. Opposite to many other bacteria, *Mtb* is capable of simultaneously consuming multiple carbon sources to augment growth, and this co-catabolism is compartmentalized for each carbon source through the glycolytic, pentose phosphate, and/or tricarboxylic acid pathways to distinct metabolic fate [38].

Here, we report for the first time the involvement of the PPE protein in sugar transport in mycobacteria. The mutation or downregulation of PPE51 in *Mtb* affected the uptake of disaccharides by tubercle bacilli. The growth of *ppe51 Mtb* mutants was also significantly attenuated in minimal media with disaccharides as the sole carbon source.

## 2. Materials and Methods

### 2.1. Bacterial Strains and Growth Conditions

*Mtb* H_37_Rv was grown at 37 °C, in Middlebrook 7H10 (Difco, Baltimore, MD, USA) medium supplemented with 10% OADC (oleic acid-albumin-dextrose-catalase). The liquid cultures were grown in Middlebrook 7H9 broth (Difco, Baltimore, MD, USA) supplemented with OADC and 0.05% Tween-80 (pH = 7). *Escherichia coli* strains were grown in Luria-Bertani broth (LB) or agar plates supplemented with ampicillin (100 μg/mL), kanamycin (50 μg/mL) or 0.01% gentamycin, if required.

To determine the MIC_90/50_ for *Mtb*, the bacilli were grown in 7H9 medium supplemented with 0.5% Tween-80 and 10% OADC. Cultures were initiated at an optical density of 0.1 (OD_600_) and supplemented with the tested chemical agents, at concentrations ranging from 1 to 0.01 mM. All *thio*-functionalized carbohydrate derivatives used in this work were synthesized and analyzed as described previously (see Appendix A) [39]. The viability assay was performed at 0, 24, 48, 72 and 96 h time points. The optical density was measured, and serial dilutions of cells were plated on 7H10 agar plates supplemented with OADC. Colonies were grown at 37 °C and counted after 4 weeks. The viability was calculated as colony forming units per mL. Experiments were performed at least in triplicate.

All single carbon source experiments were conducted in minimal media supplemented with 0.1% glycerol, 0.5% glucose, 0.5% maltose and 0.5% lactose, at pH = 5.7 and 7.0, as previously described [40]. For growth experiments, cultures were initiated at an optical density of 0.1 (OD_600_) and incubated at 37 °C, and the optical density was determined at each time point.

### 2.2. Selection of Mutants and Determination of Mutation Frequency 

The mutants resistant to T-6 disaccharide were selected on 7H10/OADC solid media supplemented with 100 µM of T-6 compound. The *Mtb* culture was (or not) preincubated in the sub-inhibitory concentration of T6 (10 µM) before plating.

To determine the frequency of T-6 resistance, the culture of *Mtb* was grown at 37 °C, until reaching an OD_600_ = 1.0 and spun 2500× *g* for 10 min, at 4 °C. Further, the bacterial cells were resuspended in 250 μL of 7H9/OADC/tween/glycerol and plated on 7H10/OADC/tween/glycerol plates supplemented with 100 μM of T6.

### 2.3. Gene Cloning Strategies

Standard molecular biology protocols were used for all cloning strategies [41]. All PCR products were obtained by using thermostable AccuPrime Pfx DNA polymerase (Invitrogen, Carlsbad, CA, USA), initially cloned into a blunt vector (pJET1.2; Thermo Fisher, Vilnius, Lithuania), sequenced and then released by digestion with appropriate restriction enzymes before ligation into the final vectors. All *ppe51* genes (with and without mutation) and their putative promoters (694-bp upstream region) were PCR amplified and cloned into the XbaI/HindIII restriction sites of the pMV306Km integration vector carrying an *attP* sequence and the integrase gene, allowing for integration of the whole plasmid into the single *attB* site in the *Mtb* chromosome. The integration was confirmed by PCR analysis. The complemented mutants were evaluated with respect to their resistance to the tested compounds. All plasmids and oligonucleotides used in this work are listed in Appendix A, respectively.

### 2.4. Next-Generation Sequencing

The sequencing libraries were prepared, using the Nextera XT DNA sample preparation protocol (Illumina Inc, San Diego, CA, USA). A total of 1 ng of genomic DNA isolated from the wild-type and 6 individual *Mtb* mutants was used for preparation of paired-end libraries, according to the manufacturer’s instructions. Whole-genome shotgun sequencing was performed on a NextSeq 500 platform, at a read length of 2 × 150 bp (300 cycles). In silico/bioinformatical analysis was performed in CLC Biology Workbench 8.0 and 8.5.1 (Qiagen). Raw sequencing reads were subjected to a quality check and adapter trimming step and further aligned to the *Mtb* H37Rv reference sequence (NC_000926) (length fraction = 0.5, similarity fraction = 0.8). Detection of SNP variants in sequencing data derived from in vitro reared resistant strains (carrying mutations) was performed, using the Basic Variant Detection algorithm. The results of variant calling for each sample were filtered against the *Mtb* H37Rv wild-type starting strain (count 10<). Low quality (quality <10) variants were excluded from further analysis.

### 2.5. Labeling of Thio-Disaccharide by Tritiation

The procedure was performed under anhydrous conditions, with a positive pressure of argon, in a fume hood, to prevent radioactive contamination. NaB_3_H_4_ (5 mCi, Hartmann Analytics GmbH, 15 Ci/mmol), freshly dissolved in anhydrous methanol (70 µL), was added to the disaccharide 1,6-anhydro-3-deoxy-4-*S*-(2,3,4,6-tetra-*O*-acetyl-β-d-glucopyranosyl-d-glycero-hexopyranos-2-ulose (1 mg, 2.04 µmol), dissolved in anhydrous methanol (300 µL) and cooled to −78 °C; the mixture was then incubated for 1 h, at room temperature, without stirring. Next, a solution of acetic acid in methanol (300 µL, 1 M) was added, and the reaction was incubated for 35 min, at room temperature. Then, the solvents were removed with a gentle stream of argon in a 25–30 °C water bath. A 1 µL sample portion was spotted on a silica gel TLC plate. The TLC plate was developed in a solvent system containing ethyl acetate and hexane 4:1, *v/v*. The radiogram was obtained by exposure to a storage X-ray screen and documented with a Kodak Medical X-ray Processor. The crude radiolabeled compound was obtained as a white solid and used for biological, assays without purification (storage at −20 °C until use).

### 2.6. T-6 Accumulation Assay

T-6 accumulation by *Mtb* was monitored in 7H9/OADC medium in cultures of living and thermally killed (80 °C, 20 min) mycobacterial cells. Tritiated T-6 (activity 2.45 Ci/mmol) was dissolved in DMSO and added to cultures grown to OD_600_ = 0.6 at a final concentration of 2 µCi/mL. During bacterial growth, 1 mL of culture samples were taken at the indicated time intervals (see Figure 1) and centrifuged at 6000× *g* for 15 min. The cell pellets were then washed twice and resuspended in Tris-EDTA buffer, prior to mixing with OptiPhase scintillation fluid (Perkin Elmer). The mycobacterial cell-associated radioactivity was determined by liquid scintillation counting, using a 1450 Microbeta Plus Liquid Scintillation Counter (Perkin Elmer). The results are presented as counts per minute (CPM). Analysis of thermally inactivated cells was used to verify whether T-6 accumulation is a result of active uptake.

### 2.7. Protein Purification

The expression construct was made by PCR amplification of *ppe51 Mtb* and cloning the amplified product into the pMALC4e [42] and pHIS [43] expression vectors. The resulting pMALC4e-*ppe51* plasmid was transformed into *E*. *coli* Arctic Express cells for protein expression and purification. Single colonies were inoculated into 5 mL of liquid LB medium, grown overnight, and diluted 100-fold in fresh medium (500 mL) supplemented with 0.2% glucose, 100 µg/mL ampicillin and 0.01% gentamycin. After the colonies were grown to midexponential phase (OD_600_ = 0.6), cells were cooled down, and protein expression was induced with 0.4 mM IPTG (isopropyl β-D-1-thiogalactopyranoside). After overnight incubation at 4 °C, cells were harvested, sonicated, and centrifuged to separate the soluble and insoluble fractions. PPE51 protein tagged to maltose binding protein (MBP) was obtained, using affinity chromatography, by passing the cell lysate through amylose resin (Amylose Resin High Flow, Rockford, IL, USA), where the recombinant protein was washed with the column buffer and eluted with eluting buffer (20 mM Tris-HCl, 200 mM NaCl, 1 mM EDTA, 1 mM DTT and 10 mM maltose). If required, the MBP-domain was cleaved out, using an Enterokinase cleavage kit (Abcam, Cambridge, UK), and captured on amylose resin. The residual enteropeptidase left in the reaction mix was removed, using cobalt beads.

The pHIS-*ppe51* construct was transformed into *E. coli* BL21(DE3), and PPE51 protein was purified from the soluble fraction by affinity chromatography, using a column packed with Ni-charged His-Bind Resin (Novagen, San Diego, CA, USA) and 6 M urea buffer, as described previously [44].

### 2.8. Production of Anti-PPE51 Rabbit Polyvalent Serum

Laboratory New Zealand rabbits were raised under standard conventional conditions approved by the Polish Ministry of Science and Higher Education animal facility of the Institute Microbiology, Biotechnology and Immunology, Faculty of Biology and Environmental Protection, University of Lodz, and were used for the immunization experiments with HIS-PPE51 recombinant protein. The experimental procedures were approved and conducted according to guidelines of the appropriate Polish Local Ethics Commission for Experiments on Animals No. 9 in Lodz (Agreement 9/ŁB87/2018). Briefly, the animals were immunized subcutaneously with three doses of recombinant PPE51 (dose I, 250 µg; doses II and III, 200 µg in 0.5 mL of PBS) emulsified with an equal volume of IFA (Incomplete Freund’s Adjuvant) in 3-week intervals. Seven days after the last booster, a blood sample was collected from the marginal ear vein, and the serum was prepared to determine the effectiveness of the immunization procedure based on the development of PPE51-specific IgG immunoglobulins detected with indirect ELISA assay and horseradish peroxidase (HRP)-labeled goat anti-rabbit IgG (Jackson ImmunoResearch) as secondary antibodies. The immune complexes were detected by using a 3,3’,5,5’-tetramethylbenzidine (TMB) liquid substrate system for ELISA (Sigma) according to the protocol recommended by the manufacturer. Absorbance values were measured at λ = 450 nm, with the reference reading at λ = 570 nm, using a Multiscan EX ELISA reader (Thermo Scientific). The optimal dilution of the secondary antibodies (1:2000) was determined in the preliminary titration experiments. The rabbit serum served as a source of primary antibodies and was used in two-fold dilutions ranging from 1:100 to 1:51200. After estimation of the immunization efficacy, the laboratory rabbits were euthanized, and blood was collected for serum preparation.

### 2.9. PPE51–T-6 Direct Interaction

To estimate the interaction of mycobacterial PPE51 with the T-6 compound, independent wells of 96-well plates (OptiPlate—96 HB; Perkin Elmer) were coated with 100 µL of recombinant PPE51*_Mtb_* or MBP-PPE51*_Mtb_* protein, and, as a control, recombinant NAD-dependent DNA ligase LigA*_Mtb_* [45] and MBP-MtrA*_Mtb_* at concentrations of 20 µg/mL in 0.1 M sodium carbonate buffer, pH = 9.5. The optimal concentration of the coating protein was evaluated in the preliminary experiments. After 90 min of incubation at 37 °C, the wells were extensively washed with PBS/0.05% Tween 20 (Sigma, Darmstadt, Germany) and blocked with 1% skim milk in PBS for 1 h, at room temperature. In the next step, all samples were washed with PBS/0.05% Tween 20, and 5 µCi/well [^3^H]T-6 in DMSO, corresponding to 1 µg of T-6, was applied to each sample for an additional 2 h, at room temperature. Unbound isotope-labeled T-6 was removed by intensively washing with PBS, and the amounts of the bound compound were determined by liquid scintillation counting. All samples and experiments were performed in triplicate. The wells coated with the recombinant proteins and incubated with PBS instead of [^3^H]T-6 served as negative controls. The protein-uncoated/blocked samples incubated with [^3^H]T-6 were used to calculate the background radioactivity.

### 2.10. CRISPR/Cas9

The CRISPR/Cas9 strategy, optimized by [46], was used to generate conditional *Mtb* mutants (KD mutants) with inducible depletion of PPE51 [47]. The appropriate sgRNA probes (Appendix A) carrying 20 nucleotide target sequences followed by strong (CTTCTTC) or weak (TTCCTTG) PAM sites were planned according to the published protocol and cloned into the pLJR965 plasmid, which was used to transform the wild-type *Mtb* strain via a standard electroporation procedure. The resulting Cas9- and sgRNA-expressing strains were confirmed for the presence of the Cas9 cassette by PCR amplification and tested for the efficiency of silencing by monitoring the PPE51 level of tested strains in the presence of the inducer anhydrotetracycline (100 ng/mL).

### 2.11. qRT-PCR

For quantitative real-time PCR (qRT-PCR) analysis of *ppe50/51* expression, RNA was extracted from wild-type *Mtb* growing in 7H9/OADC and mineral medium supplemented with cholesterol or glycerol, using TRIzol LS reagent (Invitrogen, Carlsbad, CA, USA) and mechanical disruption (FastPrep-24, MP Biomedicals). SuperScript III First-Strand Synthesis SuperMix (Invitrogen, Carlsbad, CA, USA) was used for reverse transcription, according to the manufacturer’s instructions. PCR was performed, using the Maxima SYBR Green qPCR Master Mix (Thermo Scientific, Vilnius, Lithuania) and a 7900HT Real-Time PCR system (Applied Biosystems). Each reaction included 50 ng of cDNA and 0.3 μM of each primer (Appendix A). The cycling protocol included initial heating to 95 °C for 10 min and 40 cycles of 95 °C (30 s), 60 °C (30 s) and 72 °C (30 s). Transcript levels were normalized to *sigA* gene expression as the internal control. The relative fold change was determined by using the double delta method (2^−ΔΔCt^).

## 3. Results

### 3.1. Mutated PPE51 Affects the Sensitivity of Mtb to Thio-Disaccharide

#### 3.1.1. Selection of T-6 Resistant Mutants

We have recently shown that *thio*-functionalized carbohydrate derivatives such as thioglycoside, thiosemicarbazone, aminothiadiazole and thiazoline present bactericidal activity against tubercle bacilli, inhibiting their growth at concentrations below 100 µM/CFU. At micromolar concentrations, these compounds were not active against Gram-negative *Escherichia coli* and Gram-positive *Staphylococcus aureus* [39]. Here, we applied the compound T-6, *thio*-disaccharide (1,6-anhydro-3-deoxy-4-*S*-(2,3,4,6-tetra-*O*-acetyl-β-d-glucopyranosyl-d-glycero-hexopyranos-2-ulose), at a concentration MIC_50_ = 50 µM/CFU, to inhibit the growth of *Mtb* and to identify proteins involved in the uptake of T-6-like compounds. The laboratory strain *Mtb* H_37_Rv was cultured in 7H9/OADC medium supplemented or not with a subinhibitory concentration (10 µM) of *thio*-disaccharide, and resistant mutants were selected on 7H10/OADC plates, with T-6 at a concentration of 100 µM. The calculated mutation frequency was 10^−7^, similar to what we recently found for benzimidazole derivatives, as well as for streptomycin [48]. The minimal inhibitory concentration of the compound T-6 was determined for the selected mutants as 10× (MIC_50_) or 7× (MIC_90_) higher in comparison to the wild-type strain (Table 1).

#### 3.1.2. *M. tuberculosis* T-6 Resistant Mutants Accumulate Mutations in *ppe51*

Six of the resistant mutants were subjected to genomic DNA isolation and sequencing, using the NGS Illumina system. Bioinformatics analyses allowed the identification of a number of point mutations in the investigated genomes (see Appendix A). The data analysis revealed that the gene affected in four of six investigated mutants was *ppe51*. The mutants carrying nonsynonymous substitution in *ppe51* were further investigated. To verify the relationship between the presence of the identified mutations in *ppe51* and resistance to the investigated *thio*-disaccharide, the genes carrying the mutations A96S, A66fs (196G-deletion, frame-shift mutation), T83K or L95P were cloned with the putative promoter of *ppe51* into the pMV306Km integration vector and introduced into the single *attB* locus of the genome of the wild-type strain, which was sensitive to *thio*-disaccharide.

Furthermore, using the same strategy, the wild-type *ppe51* gene of *Mtb* H_37_Rv was introduced into the genomes of the resistant mutants. The resultant merodiploid strains carrying both the wild-type and mutated *ppe51* genes were analyzed for their sensitivity to *thio*-disaccharide T-6 (Table 1). The trans-complementation of the mutants with the wild-type *ppe51,* which returned the sensitivity back to the wild-type level, confirmed the role of the investigated mutations in the resistance phenotype.

#### 3.1.3. PPE51 Mutations Do Not Affect *Mtb* Resistance to Various Anti-TB Drugs

Likewise, we investigated whether the mutations in PPE51 affect the sensitivity of *Mtb* to *thio*-disaccharide, exclusively, or to other *thio*-functionalized carbohydrate derivatives as well. We did not observe any effect of the *ppe51* mutations on the sensitivity of the *ppe51* mutant strains to aminothiadiazole (1,6-anhydro-3-deoxy-4-*S*-(5-amino-1,3,4-thiadiazol-2-yl)-d-glycero-hexopyranos-2-ulose) or thiazoline (1,6-anhydro-3-deoxy-4-*S*-(4,5-dihydrothiazol-2-yl-d-glycero-hexopyranos-2-ulose) (Appendix A). All *Mtb ppe51* mutants were also sensitive, at the level of the wild-type strain, to selected antituberculosis drugs, such as isoniazid, ethambutol, streptomycin, rifampicin and thiolactomycin (Table 2).

### 3.2. PPE51 Is Involved in the Uptake of Thio-Disaccharide by Tubercle Bacilli

#### 3.2.1. The Uptake of T-6 Is Affected in *Mtb ppe51* Mutants

Compound T-6 was radiolabeled by tritium [^3^H], as described in Methods, to monitor the uptake of thio-disaccharide by tubercle bacilli. The wild-type Mtb strain and the representative mutant carrying a substitution in PPE51 (A96S) were grown in 7H9/OADC medium to OD_600_ = 0.6; [^3^H]T-6 was then added to the culture, which was further incubated for 48 h. In the control experiment, tubercle bacilli were thermally killed before supplementation with [^3^H]T-6 to determine whether the uptake of thio-disaccharide is an active process. At the selected time points, the bacterial cells were spun down, washed and analyzed in a scintillation counter. A time-dependent accumulation of radiolabeled thio-disaccharide was observed for both wild-type and mutant strains; however, radiolabeled thio-disaccharide was significantly (*p* = 0.03) more abundant in the wild-type strain after 48 h of incubation but not after a shorter incubation time, such as 4 and 24 h (Figure 1A).

#### 3.2.2. PPE51 Binds to T-6

To determine the possible binding of *thio*-disaccharide to PPE51, the protein was expressed in *E. coli* and purified by affinity chromatography, as described in Methods (Appendix A). Recombinant PPE51, as well as PPE51-carrying MBP-domain and the NAD-dependent DNA ligase 6His-LigA [45], the protein involved in the DNA replication process, and MBP-MtrA [49] response regulator, were used as a coating experimental and control proteins, respectively, in binding experiments with 5 µCi (1 µg) of tritium-labeled *thio*-disaccharide ([^3^H]T-6). The intensity of T-6 binding was measured by liquid scintillation counting (Figure 1B) and was finally calculated by subtracting the average CPM estimated for proteins incubated without labeled T-6 from the CPM determined for the homologous protein (experimental or control) incubated with [^3^H]T-6. Disaccharide T-6 was able to bind to PPE51 but not to 6His-LigA; however, the binding of T-6 was also determined for both proteins containing MBP-domain (MBP-PPE51 and MBP-MtrA). The statistical analysis showed that the binding of T-6 was more effective to PPE51 purified from MBP-domain than MBP-PPE51 (*p* < 0.001) or MBP-MtrA (*p* = 0.002). 

### 3.3. Depletion of PPE51 Affects the Growth of Mtb in Medium with Disaccharides

#### 3.3.1. PPE51 Depletion Does Not Affect *Mtb* Growth in Rich Media

*Mtb* strains carrying mutations in the *ppe51* gene (A96S), as well as conditional mutants with inducible depletion of PPE51, were used to monitor the ability of the mutated strains to grow in minimal medium supplemented with a disaccharide as the sole carbon source. The CRISPR-Cas9 protocol was applied to engineer PPE51 conditional mutants, as described in Methods. The growth of the conditional *Mtb-ppe51* mutants was not impaired in rich medium supplemented, or not, with an inducer, anhydrotetracycline, aTc (Figure 2A). In the presence of aTc, PPE51 was depleted in the KD-mutant cells by at least 90%, according to Western blot analysis (Figure 2B).

#### 3.3.2. Mutation or Depletion of ppe51 Affect *Mtb* Growth on Disaccharides

The growth of the wild-type and mutant strains was analyzed in minimal medium (pH = 7.0) supplemented with glucose, glycerol, maltose or lactose (Figure 3A–D). When glucose was present in the minimal medium as the sole carbon source, we did not observe significant differences in the growth kinetics of any strain investigated (Figure 3A). Supplementation of minimal medium with glycerol promoted the growth of the CRISPR mutants in comparison to both wild-type *Mtb* and the mutant-carrying amino acid substitutions in PPE51 (Figure 3B). On the other hand, supplementation of the medium with either maltose or lactose promoted the growth of wild-type *Mtb*, and all mutants carrying mutated or depleted PPE51 grew significantly slower than the wild-type strain (Figure 3C,D). Similar to T-6, maltose and lactose are disaccharides which are formed from two units of glucose joined with an α(1→4) bond or from the condensation of galactose and glucose, forming a β(1→4) glycosidic linkage, respectively. Therefore, the A96S substitution in PPE51 or protein depletion affected the ability of *Mtb* to use disaccharides as the sole carbon source, suggesting the involvement of PPE51 in the uptake of such compounds.

Further, we evaluated the growth of a mutant carrying a substitution in PPE51 (T-6/2) and two CRISPR-Cas9 protein depletion mutants in minimal medium supplemented with a disaccharide (lactose or maltose), glucose or glycerol, at pH = 5.7. The growth of the wild-type strain and mutant T-6/2, but not the growth of PPE51-depletion mutants, was attenuated by approximately 50% in medium supplemented with glycerol or glucose (OD_600_ = 0.35 on glycerol vs. OD_600_ = 0.7 on maltose). Both at neutral pH and at pH = 5.7, the growth of all mutants was significantly inhibited whether the sole carbon source was maltose or lactose. On the other hand, the CRISPR-Cas9 mutants, but not the PPE51 A96S mutant, grew better at pH = 5.7 whether the minimal medium was supplemented with glycerol or glucose, suggesting that depletion of PPE51 reversed the growth attenuation effect of the wild-type strain at acidic pH on glycerol. In the case of the wild-type strain, H_37_Rv, growth arrest, however at a lower extent, was also observed on minimal medium supplemented with glycerol at neutral pH, and this phenotype was also reversed by PPE51 depletion.

The supplementation of acidic minimal medium (pH = 5.7) with glycerol or glucose promoted the growth of the PPE51-depleted mutants in comparison to the wild-type *Mtb*, as well as strains carrying mutated PPE51 (Figure 4A,B). However, the wild-type strain grew much better than the mutants in minimal medium (pH = 5.7) supplemented with either disaccharide (Figure 4C,D).

Taking the above into account, we found that the depletion or mutation of PPE51 in *Mtb* affects the ability of tubercle bacilli to use disaccharides as a carbon source. We also determined the expression level of *ppe51,* as well as its putative partner in the operon, *ppe50*, in *Mtb* growing in rich (7H9/OADC) or minimal medium supplemented with glycerol or cholesterol. Significant upregulation of *ppe51* and *ppe50* (*p* < 0.005) was observed when glycerol was the sole carbon source (Figure 5). The expression of *ppe50/51*, normalized to the expression of *sigA,* in the presence of glycerol was more than four-fold higher in comparison to the expression of *ppe50/51* in rich medium or minimal medium supplemented with cholesterol.

## 4. Discussion

It was recently reported that *Mtb* is sensitive to *thio*-functionalized carbohydrate derivatives, including *thio*-glycoside, *thio*-semicarbazone, aminothiazole and thiazoline [39]. Here, we found that the accumulation of mutations in the *ppe51* gene affects the sensitivity of *Mtb* mutants to *thio*-glycoside (*thio*-disaccharide, T-6), which might be reversed by the introduction of an intact *ppe51* gene into the mutants. Four of six *Mtb* mutants (T-6/2, T-6/7, T-6/10 and T-6/22) resistant to the compound T6 carried a nonsynonymous mutation in *ppe51*, as identified by NGS; however, two others (T-6/1, T-6/15) did not (Appendix A). Sequencing of the genome of mutant T-6/1 revealed the presence of a mutation in the *eccC5* gene resulting in the substitution D248E. As recently determined in *M. xenopi*, EccC5 is a part of the four core proteins of the ESX-5 complex, which is present exclusively in slow-growing mycobacteria and is responsible for the secretion of multiple substrates [29]. Disruption of the ESX-5 system in *Mtb* causes loss of PPE protein secretion, reduced cell wall integrity and strong attenuation in macrophages [32]. Ates [14] proposed that ESX-5 is responsible for the insertion of several channel- or pore-forming proteins that mediate the uptake of essential nutrients. Taking the above into account, the substitution in EccC5 identified in the T-6/1 mutant could affect the transport of PPE51 protein into the outer membrane and subsequently the sensitivity to the tested *thio*-disaccharide T-6. The other mutant, T-6/15, with no mutation in *ppe51,* contained a single-nucleotide deletion (187-delC) in the *pe19* gene. The PE19 protein was identified as regulated by the phosphate-sensing signal transduction system Pst/SenX3-RegX3 and the overexpression of *pe19* significantly sensitized *Mtb* to the cell wall and oxidative stresses, suggesting a unique role for PE19 in the permeability of the *Mtb* envelope [50,51]. We hypothesize that the observed resistance of the T-6/15 mutant to *thio*-disaccharide T-6 is due to the inactivation of *pe19*, followed by decreased permeability of the cell envelope.

We focused our research on *Mtb* mutants carrying missense mutations in PPE51. To verify whether the accumulation of mutations in *ppe51* affects the sensitivity of *Mtb* specifically to *thio*-disaccharide T-6, the sensitivity of the obtained mutants toward various *thio*-functionalized carbohydrate derivatives, as well as selected antituberculosis compounds, was verified (Appendix A). None of the *ppe51* mutants presented increased sensitivity to the investigated growth inhibitors in comparison with the wild-type *Mtb* strain, suggesting that the PPE51 mutations exclusively affect the sensitivity of *Mtb* to a specific group of chemicals. The T-6-resistant phenotype of the *ppe51* mutants could suggest the involvement of PPE51 in the uptake of disaccharide T-6-like compounds by *Mtb*. To verify this hypothesis, we labeled T-6 with tritium and monitored its uptake by the wild-type and *ppe51* mutant strains. The obtained results showed that the uptake of disaccharide T-6 was less effective in the *ppe51* mutant but only in 48 h incubation time. The expression and purification of the recombinant PPE51 protein allowed for the detection of its direct interaction with tritiated T-6. However, the specificity of this binding could be a matter of discussion since T-6 was also able to bind the control protein MtrA carrying MBP-domain as well. On the other hand, PPE51 without MBP-domain presented more efficient T-6 binding than MBP containing PPE51 or MBP–MtrA, suggesting that T-6 binds more efficiently PPE51 than MBP-domain. Please note that the molecular weight of the MBP domain is 42 kDa, which is 4 kDa more than PPE51, and the same amount of PPE51 contains twice more PPE51 molecules than MBP-PPE51 fusion protein. Nonetheless, the biological role of the binding of disaccharides by PPE51 remains to be elucidated by more direct experiments. The conditional PPE51 mutant constructed with CRISPR-Cas9 was used for phenotypic analysis. The depletion of PPE51 was monitored at the protein level, using rabbit anti-PPE51 serum. The amino acid substitutions in PPE51, as well as the depletion of >90% of protein, did not affect the growth of the *Mtb* mutants in rich medium. The growth of the PPE51 mutants was also not impaired in minimal medium supplemented with glucose. On the other hand, the wild-type *Mtb* strain grew much better than all investigated PPE51 mutants on minimal medium supplemented with either maltose or lactose. It was previously shown that, in response to hypoxia or starvation, *Mtb* enters a state of nonreplicating persistence (NRP) [52]. It was also reported that, when cultured in minimal medium supplemented with glycerol at pH = 5.7, *Mtb* exhibits the NRP phenotype. In the NRP state, bacilli assimilate and metabolize glycerol, maintain ATP pools and become tolerant to detergent stress and antituberculosis drugs, such as isoniazid and rifampicin [53]. Three distinct *Mtb ppe51* mutants were recently selected as strains that, as opposed to the wild-type strain, do not arrest their growth at acidic pH [53]. Based on this report, we evaluated the growth of a mutant carrying a substitution in PPE51 (T-6/2) and two CRISPR-Cas9 protein depletion mutants in minimal medium supplemented with a disaccharide (lactose or maltose), glucose or glycerol at pH = 5.7. We observed PPE51-dependent growth of *Mtb* on disaccharides and PPE51-dependent growth arrest of *Mtb* on minimal medium supplemented with glucose or glycerol at pH = 5.7. Since Baker and Abramovitch [53] analyzed *Mtb* PPE51 point mutants only, they speculated that amino acid substitutions may increase the growth of bacilli by modulating membrane permeability, possibly by modulating the channel size or specificity of PPE51, which may function as a porin to enhance access to glycerol at acidic pH. Similar observations made here with the use of PPE51 depletion mutants suggest that the mechanism of starvation at low pH is based on the role of the PPE51 protein as a cotransporter of H^₊^/glucose/glycerol, which uses the energy of the transmembrane electrochemical ion gradient to drive the accumulation of a substrate against its concentration gradient into the cell. In this mechanism, the binding of the cation (low pH) increases the affinity of the transporter for sugar [54]. On the other hand, the involvement of PPE51 in the uptake of disaccharides seems to be cation-independent. However, the role of PPE51 in the starvation of *Mtb* at acidic pH has to be further elucidated.

Based on the above observations, we asked whether *ppe51* and its partner in the putative operon, *ppe50*, are inducible in response to a specific carbon source. Quantitative PCR was used to determine the expression level of both genes in rich medium and minimal media supplemented with cholesterol or glycerol, and the data were normalized to the expression of *sigA*. The expression level of the investigated genes, *ppe50/51*, was upregulated five-fold in minimal medium supplemented with glycerol, compared to both rich and minimal media supplemented with cholesterol. This observation might suggest a role of PPE50/PPE51 in the uptake of glycerol or other hydrophilic compounds, at least in minimal media.

## Figures and Tables

**Figure 1 cells-09-00603-f001:**
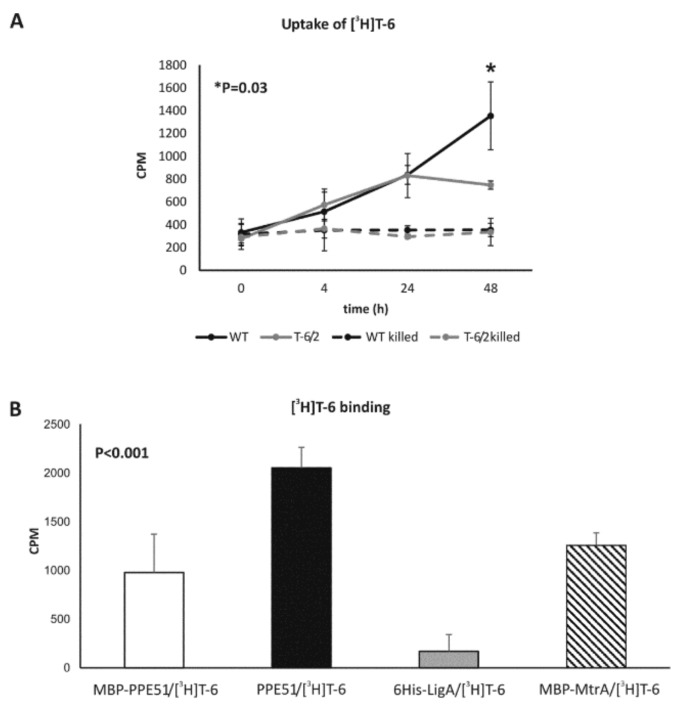
Accumulation of [3H]T-6 in *Mtb* and its binding to PPE51. (**A**) Accumulation of [3H]T-6 in living (solid line) and thermally killed (dotted line) *M. tuberculosis* wild-type (WT) and T-6/2 mutant cells. Samples were withdrawn and processed at the presented time points. The data were collected from at least three independent experiments, and the results are expressed as the means ± standard errors of the radioactivity counts per minute (CPM). (**B**) Interaction between PPE51 and tritiated T-6 (PPE51/[^3^H]T-6) expressed as the emission (CPM) of T-6 bound to the protein. As a control of binding specificity, the interactions between the PPE51 carrying MBP-domain (MBP-PPE51), response regulator MtrA carrying MBP-domain (MBP-MtrA), NAD-dependent DNA ligase (6His-LigA) and [^3^H]T-6 were analyzed. The data were collected from at least three independent experiments, and the results are expressed as the means ± standard errors. Statistical significance: (**A**) *p* = 0.03 (Student’s *t*-test), (**B**) *p* < 0.001 (one-way ANOVA, Student–Newman–Keuls test) comparing 6His-LigA to all other groups, *p* < 0.001 comparing PPE51 and MBP-PPE51, *p* < 0.002 comparing PPE51 and MBP-MtrA. Difference between MBL-PPE51 and MBL-MtrA was not statistically significant.

**Figure 2 cells-09-00603-f002:**
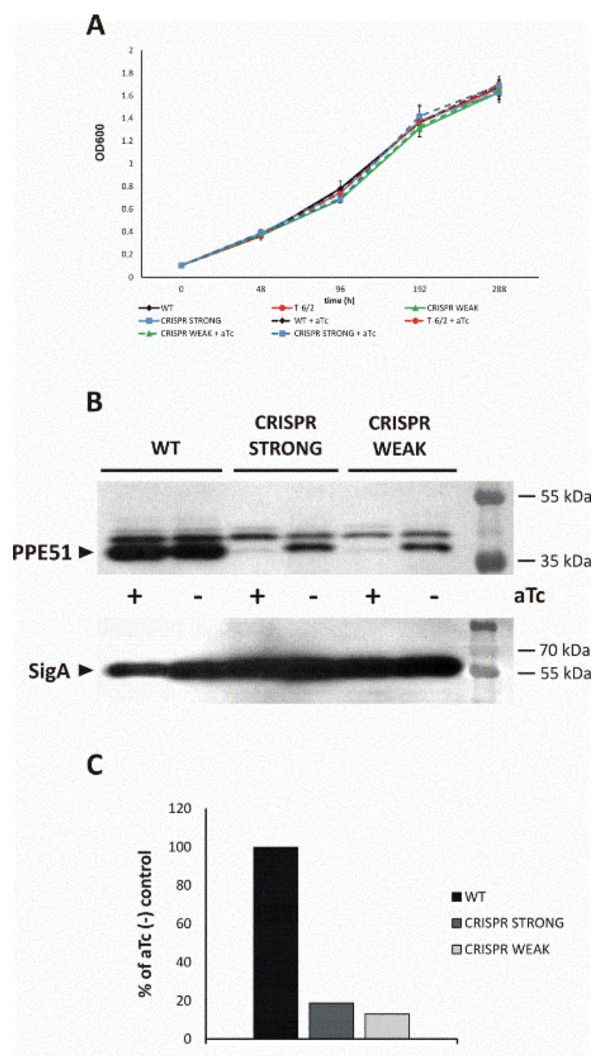
Depletion of PPE51 does not affect the growth of *Mtb* Growth kinetics (**A**) of the wild-type (black) strain and the T-6/2 (red), PPE51 CRISPR-weak (green) and PPE51 CRISPR-strong (blue) mutants of *M. tuberculosis* in 7H9/OADC medium, supplemented (dashed lines) or not (solid lines) with aTc. (**B**) Immunodetection of PPE51 and control SigA proteins in the cell lysis of wild-type *M. tuberculosis* (lines 1–2) and CRISPR mutants (CRISPR-weak, lines 3–4; CRISPR-strong, lines 5–6) grown in rich medium supplemented or not with aTc, with rabbit anti-PPE51 (upper panel) and anti-SigA (bottom panel) serum. (**C**) Densitometry analysis of bands representing PPE51 normalized to the control SigA protein.

**Figure 3 cells-09-00603-f003:**
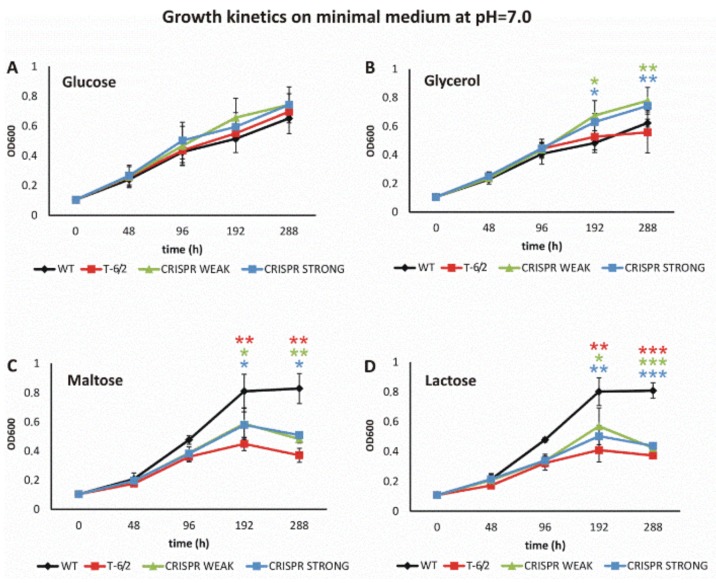
The growth kinetics of *Mtb* and *ppe51* mutants in minimal medium at pH = 7.0 Growth kinetics of the wild-type (black) strain and the T-6/2 (red), PPE51 CRISPR weak (green) and PPE51 CRISPR strong (blue) mutants of *M. tuberculosis* in minimal medium containing glucose (**A**), glycerol (**B**), maltose (**C**) or lactose (**D**), at pH = 7.0. The kinetics were evaluated by measuring the absorbance at 600 nm at the indicated time points. The displayed values are the means ± standard errors of at least three independent experiments. Statistical significance: one, two and three stars denote *p*-values ≤0.05, ≤0.01 and ≤0.001, respectively (Student’s *t*-test). Statistical significance of growth between the wild-type strain and the T-6/2 (red star), PPE51 CRISPR weak (green star) and PPE51 CRISPR strong (blue star) mutants was analyzed at 192 and 288 h.

**Figure 4 cells-09-00603-f004:**
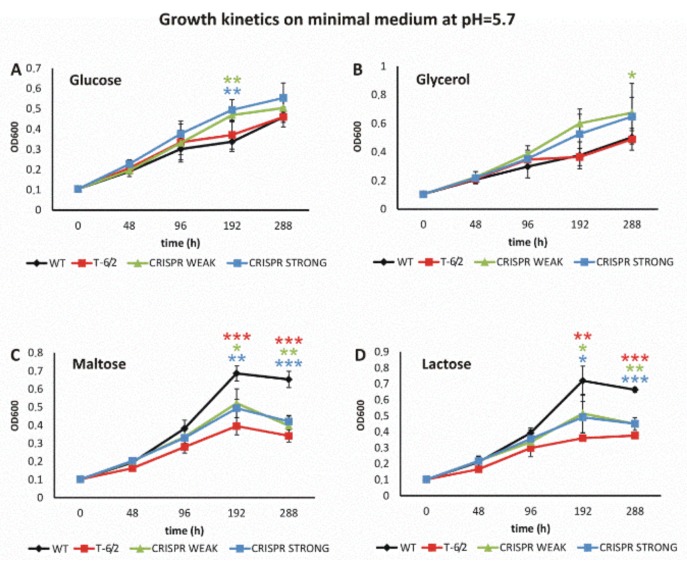
The growth kinetics of *Mtb* and *ppe51* mutants on minimal medium at pH = 5.7. Growth kinetics of the wild-type (black) strain and the T-6/2 (red), PPE51 CRISPR weak (green) and PPE51 CRISPR strong (blue) mutants of *M. tuberculosis* in minimal medium containing glucose (**A**), glycerol (**B**), maltose (**C**) or lactose (**D**), at pH = 5.7. The growth kinetics were evaluated by measuring the absorbance at 600 nm at the indicated time points. The displayed values are the means ± standard errors of at least three independent experiments. Statistical significance: one, two and three stars denote *p*-values ≤0.05, ≤0.01 and ≤0.001, respectively (Student’s *t*-test). Statistical significance between the wild-type strain and the T-6/2 (red star), PPE51 CRISPR weak (green star) and PPE51 CRISPR strong (blue star) mutants was analyzed at 192 and 288 h of growth.

**Figure 5 cells-09-00603-f005:**
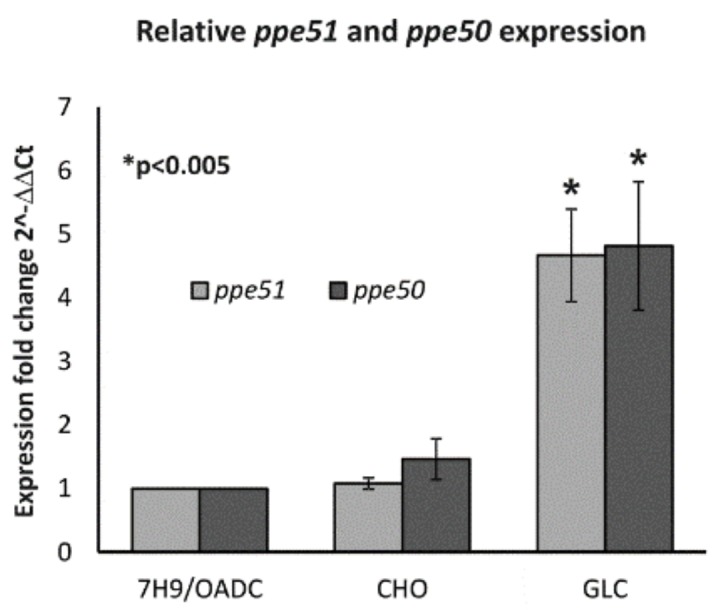
PPE51 is overproduced in minimal media supplemented with glycerol. Quantitative real-time PCR analysis of *ppe50/51* gene expression in *M. tuberculosis* growing in three different media: rich medium (7H9/OADC) and minimal medium supplemented with cholesterol (CHO) or glycerol (GLC). Transcript levels were normalized to *sigA* gene expression as the internal control. The relative fold change was determined by using the double delta method (2-^ΔΔCt^). Values are means ± standard errors Statistical significance: * *p* < 0.005 (Student’s *t*-test).

**Table 1 cells-09-00603-t001:** The minimal inhibitory concentrations of *thio*-disaccharide T-6 against wild-type *Mtb,* resistant mutants and complemented mutants.

*Thio*-Disaccharide − T-6		
Strains	MIC_50_ (µM)	MIC_90_ (µM)
H_37_Rv	50	100
**PPE51 mutants**		
T-6/2 − (A96S)	550	700
T-6/7 − (A66fs)	550	700
T-6/10 − (T83K)	550	700
T-6/22 − (L95P)	550	700
**H_37_Rv + mutated PPE51**		
H_37_Rv + (A96S) − T-6/2	50	100
H_37_Rv + (A66fs) − T-6/7	50	100
H_37_Rv + (T83K) − T-6/10	50	100
H_37_Rv + (L95P) − T-6/22	50	100
**PPE51 mutants + wild-type PPE51**		
T-6/2 + PPE51	50	100
T-6/7 + PPE51	50	100
T-6/10 + PPE51	50	100
T-6/22 + PPE51	50	100

A66fs - 196G-deletion, frame-shift mutation.

**Table 2 cells-09-00603-t002:** The bactericidal effect of selected TB drugs against wild-type *Mtb* and *ppe51* mutants.

Strains	INH	EMB	RIF	STR	TLM
	MIC_90_ (µM)				
**Rv**	**0.365**	**7.2**	**0.6**	**1.4**	**119**
PPE51 (A96S) − T-6/2	0.365	7.2	0.6	1.4	119
PPE51 (A66fs) − T-6/7	0.365	7.2	0.6	1.4	119
PPE51 (T83K) − T-6/10	0.365	7.2	0.6	1.4	119
PPE51 (L95P) − T-6/22	0.365	7.2	0.6	1.4	119

A66fs − 196G-deletion, frame-shift mutation; Isoniazid, INH; ethambutol, EMB; rifampicin, RIF; streptomycin, STR; thiolactomycin, TLM.

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
