# Peer review of "PPE51 Is Involved in the Uptake of Disaccharides by Mycobacterium tuberculosis"

_cells, 2020, doi:10.3390/cells9030603_

Round 1
Reviewer 1 Report
The manuscript of Korycka-Machala describes a set of experiments to show that PPE51 of M. tuberculosis could be involved in the uptake of disaccharides.
Overall, the findings are highly interesting and largely solid. PPE proteins have been suspected to be involved in substrate transport, but the final proof has thus far been lacking. This seems to be the most solid data at this point. As such, this work is interesting for a large audience and the presented data are novel and most of them compelling and convincing.
Having said this, there are some small issues that should be tackled.
One major point is the experiments shown in Figure 1B. Here the authors indicate that PPE51 directly binds tritiated T-6. However, they do not describe what protein is used for this? If they use the MalPPE51 fusion protein I am not surprised, because the Mal domain binds disaccharides.....Furthermore they should use as a control a protein that is relatively close to PPE51 and not ligase, i.e. another PPE protein preferably with the same Mal and/or His fusion. If the result still stands after all of this, it will have to be discussed in more detail, because this will then be a strong and very surprising indication that in fact PPE51 acts as a receptor instead of a pore.....
It is good that the authors have generated an antiserum against PPE51, but I am surprised that they hardly used it? I would suggest to test the expression of PPE51 in all mutants, inclduing the one that does not seem to have a PPE51 mutation, as it could tell something about the observed phenotype. Also for the last experiments (i,e, PE51 induction), the antiserum will be much more reliable than RT-PCR... Finally, with this antiserum in hand the authors should study where PPE51 is located. If it plays a role in transport one would expect it to be localized in the cell envelope..
Small points:
I think the data in Figure s5 is important and should be in the main document clearly mention and explain in Figure 1A that uptake is not different for the first 24 hours (and why are the other mutants not included?) Line 50, the original article that looked at EspG5 and specificity is Daleke JBC 2012, the referred article did not study transport/secretion... Figure 2B, indicate molecular weight of the markers To appreciate Figure 4 better it would be good to move (part of) section 434-449 to the results section I find the discussion rather long and it could be focused a bit more on the major finding, i.e. PPE51 is involved in growth on disaccharides. One immediately wonders how, why and whether it has close homologues that do similar things, it would be good to speculate about thisAuthor Response
Response to Referee 1:
Dear Referee,
We would like to thank the Referee 1 for his thorough evaluation of our manuscript, useful comments and we are pleased that their overall assessment, with respect to our manuscript, is positive.
The revised version of the manuscript addresses all the points raised by the Referee.
Please find specified our answers to all comments provided by the Reviewer.
Reviewer 1:
Comments and Suggestions for Authors
The manuscript of Korycka-Machala describes a set of experiments to show that PPE51 of M. tuberculosis could be involved in the uptake of disaccharides.
Overall, the findings are highly interesting and largely solid. PPE proteins have been suspected to be involved in substrate transport, but the final proof has thus far been lacking. This seems to be the most solid data at this point. As such, this work is interesting for a large audience and the presented data are novel and most of them compelling and convincing.
We would like to thank Referee 1 for a thorough evaluation of our manuscript, useful comments and we are pleased that the overall assessment, concerning our manuscript, is positive.
Answers to the specific questions:
Having said this, there are some small issues that should be tackled.
One major point is the experiments shown in Figure 1B. Here the authors indicate that PPE51 directly binds tritiated T-6. However, they do not describe what protein is used for this? If they use the MalPPE51 fusion protein I am not surprised, because the Mal domain binds disaccharides.....Furthermore, they should use as a control a protein that is relatively close to PPE51 and not ligase, i.e. another PPE protein preferably with the same Mal and/or His fusion. If the result still stands after all of this, it will have to be discussed in more detail, because this will then be a strong and very surprising indication that in fact, PPE51 acts as a receptor instead of a pore.....
We are grateful to Referee 1 for this comment. We have cloned and expressed PPE51 with 6-His and Mal tags in pHIS (pET derivative) and pMAL-c4e E.coli expression vectors, respectively. Unfortunately, the 6-HIS-PPE51 was not soluble so we used Mal-PPE51 in the binding experiments. We agree with Referee 1 that the control we used (6-HIS-LigA) does not exclude the possibility that T-6 binds to Mal(MBL) domain not to PPE51 protein. To verify this possibility, we used enterokinase treated Mal-PPE51 (Enterokinase cleavage kit, Abcam) protein which was purified further from the MBL-tag using the amylose resin and from the enzyme using cobalt beads (see Methods, 2.7). The cleavage step was very efficient however the purification from MBL(MAL) was not complete. So, we repeated the binding experiment using PPE51 (treated with enterokinase and purified), MBL-PPE51 and as a control 6HIS-LigA and additionally MBL-MtrA protein. 6HIS-LigA was the only protein we did not detect any binding of T-6. So, the specificity of T-6-PPE51 binding could be a matter of discussion since T-6 was also able to bind the control protein MtrA carrying MBP-domain as well. On the other hand, PPE51 without MBP-domain presented more efficient T-6 binding than MBP containing PPE51 or MBP-MtrA suggesting that T-6 binds more efficiently PPE51 than MBP-domain. Please note that the molecular weight of the MBP domain is 42 kDa which is 4 kDa more than PPE51 and the same amount of PPE51 contains twice more PPE51 molecules than MBP-PPE51 fusion protein. Nonetheless, the biological role, if any, of the binding of disaccharides by PPE51 remains to be elucidated by more direct experiments.
We have corrected the Methods (section 2.7) by including the information about enterokinase cleavage and purification. Section 2.9 describing the binding experiment has been corrected. Results (section 3.2.2 and Fig. 1) were corrected by introducing additional control data and statistical analysis. We have discussed more critically the obtained results about T-6-PPE51 binding in the discussion chapter (lines 527-535).
To downgrade the role of T-6-PPE51 binding in this story, we have removed a sentence mentioning it from the abstract and from the Introduction.
We agree with Referee that it would be good to know whether 3HT-6 binds to other PPE proteins, however, such protein was not available in the lab, and cloning/purification would exceed the available time for the submission of the revised version of the manuscript.
Even we see some in vitro PPE51 and 3HT-6 interaction, we cannot say that PPE51 is a receptor for di-saccharides. Further experiments would be required to assess the strength of the interaction (SPR and/or thermophoresis) and to visualize the interactions in situ. Such studies exceed the story described in this manuscript. Based on the experiments we made, we are convinced that PPE51 is involved in the uptake of disaccharides, but we rather think that the protein work rather as a pore as suggested by Referee, even it is able to bind the sugar.
It is good that the authors have generated an antiserum against PPE51, but I am surprised that they hardly used it? I would suggest to test the expression of PPE51 in all mutants, inclduing the one that does not seem to have a PPE51 mutation, as it could tell something about the observed phenotype. Also for the last experiments (i,e, PE51 induction), the antiserum will be much more reliable than RT-PCR... Finally, with this antiserum in hand the authors should study where PPE51 is located. If it plays a role in transport one would expect it to be localized in the cell envelope..
We generated antiserum mainly to monitor the protein depletion in CRISPR-Cas9 mutants and to use it in an affirmative experiment of PPE51 and 3HT-6 interaction. We purified PPE51 Abs to capture PPE51 protein from Mtb (wild type and mutants) cell lysates enriched with 3HT-6. Unfortunately, validating the protocol of the experiment, we found that 3HT-6 binds heavily to the control Abs, so we could not distinguish between the binding to Abs and binding to Abs-PPE51, so data was not included in the manuscript.
We did not control the level of PPE51 in mutants carrying point mutations in ppe51 because we did not suppose to find it as very informative. The strain carrying ppe51-missense mutation would still produce a new protein with PPE51 N-terminal part which would be recognized by Abs. We did not expect that the substitutions in PPE51 of other mutants could affect the PPE51 level. It’s also likely that mutated eccC5 (Esx5) might affect rather the localization of PPE51 than its expression.
We found that the expression of ppe51 is induced on glycerol vs cholesterol or rich medium. In our opinion, the qPCR is more reliable than western blot to quantitate the medium-dependent expression of any gene, including ppe51. It is easier to quantitate it (by normalizing to a constitutive gene) and statistically analyze the difference if any. Abs dependent experiments usually rise the questions about signal (band) saturation, the affinity of antibodies and when using different growth media what is the percentage of a given protein in a whole proteome.
The localization of the PPE51 with Abs would be a good point and we would like to continue the story in this direction, determining the localization of the protein, its transport, and partners if any.
Small points:
I think the data in Figure s5 is important and should be in the main document
There is no Fig. S5 in the supplementary material, so I guess Referee thought about Table S5 describing the drug-resistance analysis of mutants. We included this Table in Supplementary materials since we did not detect any effect of ppe51 mutation on resistance to the anti-TB compounds.
As suggested by the Referee, Table S5 has been translocated into the main text in the R1 version (Table 2).
clearly mention and explain in Figure 1A that uptake is not different for the first 24 hours (and why are the other mutants not included?)
The sentence was modified following the Referee suggestion. Lines 344-345
We have included only a single mutant in this experiment because the amount of available 3HT-6 was quite limited.
Line 50, the original article that looked at EspG5 and specificity is Daleke JBC 2012, the referred article did not study transport/secretion...
Thank you very much for this correction. The missed reference Daleke et al., 2012 was introduced into the text (reference 13) and reference 9 (new reference 12) was moved to the sentence describing the PE/PPE/EspG5 complex.
Figure 2B, indicate molecular weight of the markers
Corrected as suggested by Referee
To appreciate Figure 4 better it would be good to move (part of) section 434-449 to the results section
We followed the Referee’s suggestion
I find the discussion rather long and it could be focused a bit more on the major finding, i.e. PPE51 is involved in growth on disaccharides. One immediately wonders how, why and whether it has close homologues that do similar things, it would be good to speculate about this
The discussion has been significantly shortened following the suggestions of both Referees. We haven’t discussed the putative function of PPE51 homologs. We did not make such analysis and there is no data available in the literature that could support such discussion. We rather tried to avoid such speculations. Of course, after the evidence that PPE-proteins are involved in heme utilization and disaccharides, we would not be surprised to see the next papers describing the role of PPE-proteins in the uptake of other nutrients.

Reviewer 2 Report
The article "PPE51 is involved in the uptake of disaccharides by Mycobacterium tuberculosis" is of great scientific and practical interest for researchers working in the field of medical microbiology, genetics and pathogenesis of M. tuberculosis. According to the WHO Tuberculosis is one of the 13 pathogenic bacteria posing a real epidemiological threat. Therefore, the necessity to create genetically engineered vaccines and drugs with a new mechanism of action active against MDR strains can bot be emphasized enough.
The PE/PPPE family of proteins is involved in catagenesis and virulence mechanisms at various stages of M. tuberculosis infection. However, due to the complexity and heterogeneity of this network, the functions of the proteins of this family are poorly understood. It is known that these proteins are, just like antigens, possibly involved in the development of tolerance and anti-TB antibiotics. For this reason, establishing the role of the PPE51 protein in the transport of certain disaccharides is of utmost importance.
I have comments on each section of the article.
Introduction. Please include some information on the spread of tuberculosis, drug resistance, virulence, pathogenicity and the role of the PE / PPPE family in this. Also expand on what is known about the use of sources of carbohydrates and other carbonates in M. tuberculosis.
Materials and methods. Please give a description of the hio-disaccharides used in the work. Please give a description of the procedure of producing stable mutants, selection criteria, stability of phenotype conservation.
Results.
Please transfer the description of thio-functionalized carbohydrate into the materials and methods section. Please describe in more detail the methods: quantity, morphological physiological characteristic, possible distribution into groups. Why did the authors, before plating the culture on a solid medium supplemented with the inhibitor, grow it in a liquid medium supplemented with subinhibitory concentrations of this inducer of resistance mechanisms? As it is known, mutations are a global impediment in the fight against MDR/XDR M. tuberculosis strains. As a rule, many publications pay little attention to this. If the authors had received more mutations instead of just six, it would have been preferable to analyze these PCP clones for the presence of any already established mutations. Overall, section 3 should be divided into several subsections 3.1.1., 3.1.2, 3.1.3.
I recommend the subdivision of section 3.3. into subsection 3.3.1. Development of the technology of creating mutants with a modified promoter region using CRISPR-Cas9. The use of CRISPR-Cas9 technology is instrumental in the study of the properties and specific genes of M. tuberculosis including virulence, pathogenicity of MDR / XDR resistance.
The advantage of the presented work is its interdisciplinarity and the use of a complex of modern methods, technologies, approaches, and these aspects need to be emphasized more.
Discussion Section. It is necessary to remove the first section dedicated to the characterization of mutants T-6/1, T-6/15, which are not related to the title of the work and further mentioned in the result section. In short, this section falls outside the general scope of this study.
Author Response
Response to Referee 2:
Dear Referee,
We would like to thank the Referee 2 for his thorough evaluation of our manuscript, useful comments and we are pleased that their overall assessment, in respect to our manuscript, is positive.
The revised version of the manuscript addresses all the points raised by the Referee.
Please find specified our answers to all comments provided by Reviewer.
Reviewer 2:
Comments and Suggestions for Authors
The article "PPE51 is involved in the uptake of disaccharides by Mycobacterium tuberculosis" is of great scientific and practical interest for researchers working in the field of medical microbiology, genetics and pathogenesis of M. tuberculosis. According to the WHO Tuberculosis is one of the 13 pathogenic bacteria posing a real epidemiological threat. Therefore, the necessity to create genetically engineered vaccines and drugs with a new mechanism of action active against MDR strains can bot be emphasized enough.
The PE/PPPE family of proteins is involved in catagenesis and virulence mechanisms at various stages of M. tuberculosis infection. However, due to the complexity and heterogeneity of this network, the functions of the proteins of this family are poorly understood. It is known that these proteins are, just like antigens, possibly involved in the development of tolerance and anti-TB antibiotics. For this reason, establishing the role of the PPE51 protein in the transport of certain disaccharides is of utmost importance.
We would like to thank Referee 2 for a thorough evaluation of our manuscript, useful comments and we are pleased that the overall assessment, concerning our manuscript, is positive.
Answers to the specific questions:
I have comments on each section of the article.
Introduction. Please include some information on the spread of tuberculosis, drug resistance, virulence, pathogenicity and the role of the PE / PPPE family in this. Also expand on what is known about the use of sources of carbohydrates and other carbonates in M. tuberculosis.
Following the Referee’s recommendation, we introduced some basic-TB information into the text of Introduction (references 1-3).
We have already included the information about the role of PE/PPE in the pathogenicity of Mtb (lines 74-79) where we cited a number of papers (citations 19-25) describing this problem. To help readers, we have introduced one more citation in this field, a nice review paper (reference 19).
The short information about the carbon sources available during infection of Mtb was also introduced into the text of Introduction (lines 91-98).
Materials and methods. Please give a description of the hio-disaccharides used in the work. Please give a description of the procedure of producing stable mutants, selection criteria, stability of phenotype conservation.
The thio-disaccharides used in this work were described in detail (including synthesis) in reference 30 (R1 citation 39) cited in chapter 2.1. To help readers to follow the story we additionally introduced a Table (new Table S1) carrying the basic information about all compounds used.
The CRISPR-Cas9 mutants were generated as described in detail in paragraph 2.10, and cited references (new 37, 38 – new R1-46, 47) as well as Table S2 (new Table S3).
We have introduced into the Methods a new subsection (new 2.2) describing in detail the selection of T-6 resistant mutants and determination of the frequency of mutation.
In every experiment, the mutants (point mutations) were verified in respect to their resistance-phenotype to T-6 and by PCR/sequencing to confirm the presence of the mutation. The phenotype of CRISPR-Cas9 mutants was controlled at the PPE51 protein level (inducible depletion of PPE51 in the presence of aTc). The stably maintaining of CRISPR-Cas9 integrative plasmid was enforced by the antibiotic pressure.
Results.
Please transfer the description of thio-functionalized carbohydrate into the materials and methods section.
There are two sentences at the beginning of the Results section describing thio-saccharides used in this story. I would prefer to keep it in the Results since it helps readers to understand the strategy of our investigations.
Please describe in more detail the methods: quantity, morphological physiological characteristic, possible distribution into groups.
The methodology was enriched by the description of the selection of T-6 resistant mutants and the determination of the frequency of mutation.
Why did the authors, before plating the culture on a solid medium supplemented with the inhibitor, grow it in a liquid medium supplemented with subinhibitory concentrations of this inducer of resistance mechanisms?
We used both, direct selection of mutants on plates supplemented with a high concentration of T-6 as well as preselection using the sub-inhibitory concentration of T-6 followed by the selection of T-6 resistant mutants. Of course, to determine the frequency of mutations we did not make the preselection. We supposed that the pre-incubation of Mtb in the sub-inhibitory concentrations of T-6 might increase the mutation rate (e.g. inducing SOS) to make the selection easier. We haven’t seen any difference in frequency using or not preselection, likely because the SOS response is not induced in the presence of sub-inhibitory concentrations of T-6.
As it is known, mutations are a global impediment in the fight against MDR/XDR M. tuberculosis strains. As a rule, many publications pay little attention to this. If the authors had received more mutations instead of just six, it would have been preferable to analyze these PCP clones for the presence of any already established mutations.
We have selected many mutants, but six of them were subjected to NGS and analyzed in detail.
Overall, section 3 should be divided into several subsections 3.1.1., 3.1.2, 3.1.3.
We have introduced subsections as suggested by the Referee.
I recommend the subdivision of section 3.3. into subsection 3.3.1. Development of the technology of creating mutants with a modified promoter region using CRISPR-Cas9. The use of CRISPR-Cas9 technology is instrumental in the study of the properties and specific genes of M. tuberculosis including virulence, pathogenicity of MDR / XDR resistance.
The CRISPR-Cas9 technology we used was described in detail in the Methods section, we also cited the original paper containing vectors, PAM sequences, and procedure how to plan, make and use this technique in fast and slow-growing mycobacteria. We could not make the subsection “Development of the technology …” because the technology was developed already by Sara Fortune group and kindly shared with us.
The advantage of the presented work is its interdisciplinarity and the use of a complex of modern methods, technologies, approaches, and these aspects need to be emphasized more.
Thank you very much for this opinion, we did our best to improve this by modifying the Results section (Please see the corrections made in response to Referee 1).
Discussion Section. It is necessary to remove the first section dedicated to the characterization of mutants T-6/1, T-6/15, which are not related to the title of the work and further mentioned in the result section. In short, this section falls outside the general scope of this study.
We have sequenced 6 mutants resistant to T-6 by NGS and 4 of them carried the mutations in PPE51. The complementation assay showed the direct relationship between ppe51 mutation and resistance to T-6. However, it raised the question about 2 other mutants. So we were obliged to explain how the mutants without ppe51 mutation developed the resistance to T-6. This part of the discussion was abbreviated.
Please note, that part of the Discussion chapter was moved into the Results section as suggested by Referee 1.
